# Association between Leucocyte Telomere Length and Risk of Hearing Loss in the General Population: A Case-Control Study in Zhejiang Province, China

**DOI:** 10.3390/ijerph17061881

**Published:** 2020-03-13

**Authors:** Huai Zhang, Dahui Wang, Haiyan Ma, Chenhui Li, Shichang Wang, Yi Wang, Lei Yang, Liangwen Xu

**Affiliations:** The Medical School, Hangzhou Normal University, Hangzhou 311121, China; zhanghuaizhanghuai@163.com (H.Z.);

**Keywords:** hearing loss, relative telomere length, case-control

## Abstract

Limited studies have assessed the relation between telomere length and risk of hearing loss; moreover, they have reported equivocal associations. In the first case-control study, the subjects were chosen from the general population of Zhejiang province in order to assess the association between leucocyte telomere length and risk of hearing loss from 2016 to 2018. A total of 817 cases (55.93 ± 8.99 years) and 817 age-, sex- and residential city-matched controls (55.91 ± 9.03 years) were included for analysis. In the multivariable models, individuals in the top quartile of relative telomere length (RTL) had an odds ratio (OR) for hearing loss of 0.53 (95% confidence intervals [CI], 0.38–0.74) compared to those in the bottom quartile, and specifically, the OR was 0.45 (95% CI, 0.28–0.73) in females. In females, the risk of hearing loss decreased by 46% as RTL doubling increased; the standard deviation of RTL was associated with a 29% decrease in hearing loss risk. Additional analysis showed significant difference between participants in the female mild hearing loss group and corresponding controls. These results suggest that telomere length is associated with hearing loss in the general population, particularly in females with mild hearing loss. Telomere length might be a potential predictive biomarker of hearing loss at early stage.

## 1. Introduction

Hearing loss, a disease included in the ear, nose, and throat medical specialty, has become the most common sensory disorder in humans [1]. Data from different research institutions indicate that the incidence of hearing loss has increased in recent decades, and therefore, hearing loss is increasingly gaining attention. A retrospective study published in the Lancet journal reported that over the past 26 years, hearing loss has been one of the top-ranked diseases among 328 diseases and injures in 195 countries in terms of both prevalence rate and years lived with disability (YLD) [2]. The World Health Organization also reported an increase in the prevalence of hearing loss from 42 million in 1985 to 360 million in 2011 globally [3]. Moreover, the enormous direct and indirect economic burden caused by hearing loss warrants prompt attention in view of the prevalence of a large number of cases [4]. It appears increasingly likely that hearing loss is a lifetime problem with childhood origins [5], with its prevalence increasing with age [6]. However, the exact mechanisms of hearing loss, particularly the biological mechanisms, have not been fully identified, although genetics, effects of ototoxic drugs and noise exposure have been the most-studied risk factors.

Telomeres, repetitive DNA sequences (TTAGGG repeats) extending from a few bases to 20 kilobases in length [7,8], are specialized structures located at the ends of eukaryotic chromosomes and play crucial roles in the maintenance of chromosome stability and integrity [9]. Telomeres shorten by 30–200 base pairs annually in human somatic cells, in vivo, until they reach the Hayflick limit [10,11]. Although large interlaboratory differences exist in the values of annual attrition of telomeres [12], undoubtedly, most human somatic cells or tissues experience irreversible telomere shortening with age [7]. Therefore, the telomere may be a useful biomarker for studying and predicting the onset of age-related diseases, such as cardiovascular disease, cancer, and mortality [13]. Moreover, this may identify the relationships between telomere shortening and inflammation, reactive oxygen species (ROS), and endothelial damage, all of which may contribute to hearing loss [14]. 

However, limited evidence exists in domestic and foreign databases [15,16,17] that directly examines the relationship between hearing loss and telomere length, and moreover, the available literature has certain limitations. For example, these investigations include either specific populations (specific occupational workers or elderly adults) or specific types of hearing loss (presbycusis or sudden sensorineural hearing loss); additionally, the results reported are inconsistent. Furthermore, the telomere length, unlike biochemical indices, such as routine blood and plasma lipid examinations, does not have large fluctuations in a short time. That is, telomere length has a certain stability. Therefore, in this first case-control study assessing hearing loss and telomere length, we collected the data from audiometric measurements, responses to structured questionnaires, and laboratory examinations to study the correlation between telomere length and hearing loss in the general population. These study results may be of possible significance for the in-depth understanding of mechanism of hearing loss, early detection of hearing loss, assessing the effect of telomere length on hearing loss, and for the development of new strategies to treat patients with hearing loss. 

## 2. Methods

### 2.1. Study Design and Participants

A case-control study using a multistage stratified cluster random sampling method was conducted in the Zhejiang province from 2016 to 2018. Data from the following seven healthcare centers in Zhejiang was collected: one center in Quzhou (Jiangshan city), one in Jiaxing, one in Huzhou, one in Lishui, and three in Hangzhou (Tonglu county; Baiyang and Sijiqing communities). Thereafter, we randomly selected 1008 individuals with hearing loss, and randomly matched each control with each case by age (±1 year), sex, and residential city. Finally, 1008 controls were selected for this study; all subjects agreed to provide the relevant information and consented to its use in future research. Furthermore, given that prevalence of hearing loss increases with age, and that shortening of telomere length is believed to be coupled to the aging process, we selected a study population comprising only middle-age and older individuals (>40 years).

The study was approved by the Institutional Review Board of Hangzhou Normal University (grant number, 2017LL107). All subjects provided written informed consent and the entire study was performed in accordance with the ethical standards laid down in the 1964 Declaration of Helsinki and its later amendments, and according to local government policies.

### 2.2. Measurement of RTL in Leucocytes

At the time of visiting the healthcare centers, a venous blood sample was obtained from the antecubital vein of each patient for the extraction of genomic DNA [18]. Genomic DNA was extracted from the peripheral blood cells using standard procedures described in the TIANamp Genomic DNA Kit (Tiangen Biotech, Beijing, China). The extracted DNA was quantified using a Nanodrop 2000 spectrophotometer (Thermo Scientific, Waltham, MA, USA) and was stored at −80 °C until the time of telomere length measurement. Participants whose blood samples gave poor quality of extracted DNA (*n* = 11; i.e., OD (optical density) 260/OD280 < 1.60 or OD260/OD280 > 2.00) were excluded. After adjusting the DNA concentration of all samples to 10 ng/uL with 10 mmol/L Tris-ethylenediaminetetraacetic acid buffer solution (pH = 8.0), real-time quantitative-polymerase chain reaction (RT-qPCR) was performed to measure the relative telomere length (RTL), using the ratio of the telomere repeat copy number with the copy number of the b-globin (single copy gene) [19,20]. The RT-qPCR reactions were performed in duplicate for each sample in a PikoReal 96 Real-time PCR System (Thermo Scientific). Each 10 µL of the RT-qPCR reaction had 5 µL of SuperReal PreMix Plus (2×) and the following primers for measuring the telomere length: Telomere forward primer, 5′- CGG TTT GTT TGG GTT TGG GTT TGG GTT TGG GTT TGG GTT -3′; Telomere reverse primer, 5′- GGC TTG CCT TAC CCT TAC CCT TAC CCT TAC CCT TAC CCT -3′; b-globin gene forward primer: 5′- GCT TCT GAC ACA ACT GTG TTC ACT AGC -3′; b-globin reverse primer: 5′- CAC CAA CTT CAT CCA CGT TCA CC -3′. The thermal cycling profiles for both the genes started with a 95 °C pre-degeneration step for 15 min, followed by 40 cycles of 10 s at 95 °C and 31 s at 60 °C. Reference DNA sample from Hela cells, serially diluted from 25 to 1.5625 ng/µL (2-fold dilution; five data points), were used to generate the standard curve for each of the 96-well PCR plates. For each sample, the number of telomere repeats and b-globin copies were determined in comparison to the reference sample [18]. The inter- and intra-plate coefficient of variation ranged from 6%–15% in our study for the RTL measurement [21].

### 2.3. Audiometry and Other Covariates

Participants with pre-existing ear diseases (such as externa, otitis media, or cerumen impaction) or abnormal ear structure were excluded by otoscopy (*n* = 51). The previously excluded population (*n* = 11, the previously excluded population, and the present *n* = 51 are not mutually exclusive) was combined with the present number of excluded participants for the assessment of study characteristics, and only the middle-age and elderly individuals were included (*n* = 857); hence, a total of 1634 participants (817 cases and 817 controls) were eventually included for analysis. All pure-tone air-conduction hearing thresholds were measured by trained researchers using audiometers (AT235; Interacoustics AS, Copenhagen, Denmark) with supra-aural headphones (TDH-39; Telephonic Corporation, Farmingdale, USA). Each participant was evaluated for hearing thresholds between 0.125 and 8 kHz (0.125, 0.25, 0.5, 1, 2, 3, 4, 6, and 8 kHz) in a soundproof booth with background noise of less than 20 dB (A). A detailed protocol for audiometry has been described previously [22]. In this study, hearing loss was defined as the pure-tone average (PTA) of the speech frequencies (0.5, 1, 2, and 4 kHz) of ≥26 dB in either ear [23,24]. Furthermore, the degrees of hearing loss were classified according to the following PTAs: normal (≤25 dB), mild (26–40 dB), moderate (41–60 dB), severe (61–80 dB), and extremely severe (≥81 dB) [25]. 

A structured questionnaire for collecting data on the demographic variables, audiometric information, and issues related to various risk factors and diseases was completed by each participant in the presence of a healthcare official. Self-reported medical information, mainly on hypertension, hyperlipidemia, diabetes, and hypercholesterolemia, was collected additionally. Moreover, blood levels of lead and cadmium were also detected using graphite furnace atomic absorption spectrometry (BH2200S; Beijing Bohui Innovation Biotech; Beijing, China). Compared with external factors, such as smoking and occupational exposure (such as battery manufacturer workers), the blood levels (internal doses) of lead and cadmium could better reflect the in vivo conditions. 

### 2.4. Statistical Analyses

An estimated 776 cases and 776 controls would be needed to provide 90% power for the present case-control study, assuming an odds ratio (OR) of 0.67 when comparing long RTL with short RTL (corresponding to a median RTL of 1.20 vs. 0.73 for long and short RTL, respectively), with a two-sided α of 0.05. In this study, exceeding the statistical requirement, we analyzed the data of 817 cases and 817 controls to ensure sufficient power of the study and statistically relevant results. Epidata V.3.1 (The Epidata Association, Odense, Denmark) was used for all raw data entry, checking, and error correction (double entry and validation). All statistical analyses were conducted using the SPSS V.19.0 software (SPSS Inc., Chicago, IL, USA), and the results were plotted using Adobe Illustrator CS5 software (Adobe Systems Inc., San Jose, CA, USA). The Kolmogorov-Smirnov normality test was conducted to examine the distribution of each variable for presenting the data as mean (standard deviation [SD]), median (interquartile range), or proportions. Because of the case-control study design, the paired *t*-test (both lead and cadmium exhibited a right-skewed distribution and were log-transformed in this test), McNemar test, and Wilcoxon signed-rank test were used to compare differences between the controls and cases. Conditional logistic regression analysis was used to estimate the association between hearing loss and the variables. All reported probability values were two-tailed, and *p*-values less than 0.05 were considered statistically significant. 

## 3. Results

The final sample size in the pooled analysis comprised 817 cases and 817 controls (41–89 years old), who were matched for age (±1 year), sex, and residential city. The average age was 55.93 (8.99) years among the cases and 55.91 (9.03) years among the controls. Except for participant sex and lead concentration, the other variables were significantly different between the case and control groups. Notably, compared with the control group, participants with hearing loss had shorter RTL (0.92 [0.71–1.15] vs. 0.97 [0.75–1.22]; *p* = 0.004) (Table 1). 

Table 2 presents the results of the conditional logistic regression analysis; the variables with statistically significant differences in Table 1 were included in the regression analysis. We observed an inverse association between RTL and risk of hearing loss. Participants in the top quartile of RTL had an odds ratio (OR) for hearing loss of 0.53 (95% CI, 0.38–0.74) compared to those in the bottom quartile with a significant trend across the quartiles (*P* for trend = 0.002). Among the common chronic diseases, hypertension (OR = 2.10; 95% CI, 1.62–2.71) and diabetes (OR = 1.91; 95% CI, 1.08–3.38) were found to have a significant positive correlation with hearing loss. However, no significant association was found between presence of hyperlipidemia or hypercholesterolemia and hearing loss (for hyperlipidemia: OR = 1.44; 95% CI, 0.86–2.42; for hypercholesterolemia: OR = 1.34; 95% CI, 0.62–2.90). Furthermore, as the focused variable—RTL—we modeled it as a continuous variable (per doubling or per SD increase in RTL). The per SD increase values for RTL was significantly associated with a 21% decrease risk of developing hearing loss (OR = 0.79; 95% CI, 0.70–0.90), and the per doubling increase in RTL was significantly associated with a 34% decrease in the risk of hearing loss (Table 3). Further adjustment for potential confounding factors did not change the results considerably. In addition, although sex and age were matched, we still separated the population into 4 groups, to analyze the effect for hearing loss by RTL levels stratified by sex and age (60y as the critical age) (Table A1). The result showed that the effect of telomere shortening on hearing loss was more sensitive in females regardless of age. Specifically, in females, participants in the top quartile of RTL had an OR for hearing loss of 0.45 (95% CI, 0.28–0.73), as compared to those in the bottom quartile, with a significant trend across the quartiles (*P* for trend = 0.011); RTL doubling was associated with a 46% decrease in the risk of hearing loss (OR = 0.54; 95% CI, 0.38–0.76); the per SD increase values for RTL was significantly associated with a 29% decrease in the risk of hearing loss (OR = 0.71; 95% CI, 0.58–0.860).

In this study, among all the participants included (41–89 years old), the scatter plot showed that RTL negatively correlated with age (Figure A1), which also confirmed the presence of telomere shortening mentioned in the introduction section. Furthermore, as described in the method section, the population with hearing loss was divided into the following categories: mild (*n* = 543), moderate (*n* = 220), and (extremely) severe (*n* = 54, the severe and extremely severe populations were merged because of small numbers in both subgroups). As Table A1 indicates, biological sex could moderate observed links between RTL and hearing loss, and a sex-specific analysis was performed in Figure 1 to show the difference in RTL between the cases and the corresponding controls after adjustment for confounding factors and subsequent stratification by the degree of hearing loss. Compared with the participants having hearing loss, the control groups had longer RTL; in particular, a significant difference was observed in the female mild hearing loss subgroup (*p* = 0.004).

## 4. Discussion

The present province-wide, population-based case-control study identified the association between hearing loss and telomere length, and reports that longer RTL was associated with a lower risk of developing hearing loss. Furthermore, we report that the association remained significant even after adjusting for hypertension, hyperlipidemia, diabetes, hypercholesterolemia and cadmium in blood (Table 3). Moreover, we further analyzed the differences in telomere length between different subgroups of hearing loss and corresponding controls by sex-specific data. To the best of our knowledge, it is the first report on the relationship between telomere length and hearing loss in the case-control study design format. The findings of this study indicate that telomere length might be a potential predictive biomarker in hearing loss at early stage.

In vivo, repeated DNA sequences (TTAGGG) of telomere are made by the telomerase enzyme during cell division. However, telomerase activity is repressed critically in most physiological, mature somatic cells, except for germline cells, stem cells, and lymphocytes [7,26]. The limited expression of telomerase appears to be a mechanism of tumor suppression in humans [27]; moreover, the reason why the human telomere acts as a mitotic clock of replicative aging remains unknown. Hearing loss usually manifests as progressive aggravation in the general population. In an epidemiological survey, the highest prevalence of hearing loss was observed in the elderly population. Therefore, in this study, we first adjusted the confounding effects of age, sex, and geographic factors in the data collection phase with the case-control study design. Moreover, an established, widely-used quantitative PCR-based technique, which has been previously validated against terminal restriction fragment analysis with a high relationship, was used to measure RTL [19]. 

This study reports that the risk of hearing loss increases significantly in people with shorter telomeres. Subjects in the top quartile of RTL had significantly less risk of developing hearing loss as compared to those in the bottom quartile who had a relatively shorter telomere length. When modeled continuously with covariate adjustment, per doubling (or per SD [0.374]), RTL was associated with a 33% (or 18%) decrease in the risk of hearing loss. Oxidation damage, including cochlear DNA damage, caused by ROS plays a causal role in the development of hearing loss [28], and the following factors (mechanisms) [29,30] promote the occurrence of hearing loss: multiple mechanisms that can promote telomere shortening, including oxidative stress [31] and inhibition of DNA repair [32]. There is already evidence that shortened telomere can result in chromosomal instability and promote genomic lesions [7,33]. The telomere sequence is highly sensitive to damage by ROS because of the enrichment of the GGG triplet. ROS can cause single-strand breaks, whereas telomere DNA may have single-strand breaks repair defects, in contrast to the majority of the genomic DNA [32]. This may be due to the formation of 8-oxodeoxyguanine, a key oxidation product of guanine, which can lead to DNA replication mismatch. Additionally, it is also worth noting that, although hearing loss is higher in males in the general population [22], telomere shortening has a greater impact on hearing loss in females (Table A1). It indicates that those factors (RTL, chronic conditions and cadmium) affecting hearing loss do not play exactly the same role in males and females.

In particular, for the study of hearing loss and telomere length, three cross-sectional studies have assessed the association between RTL and the risk of hearing loss; two studies reported an inverse association [15,17], whereas one study reported null findings [16]. Differences in the research objectives, definitions of hearing loss, and study design and methods may be the reasons for inconsistencies between the results of different studies. We further examined the different degrees of hearing loss by being sex-specific, especially for mild hearing loss (similar to that reported by Wang et al. [16]), to identify the early associations between telomere length and lifetime risk of hearing loss. From our results (Figure 1), we believe that the reason for the inconsistencies in previous studies may be the differences in the proportions of populations with different degrees of hearing loss in these studies. As a progressive aggravating sensory disorder, telomere length was found to be significantly associated with early stages of hearing loss in females (i.e., mild hearing loss), but the association with non-mild hearing loss was not significant. We speculate that there might be an early compensatory reaction between the two, which may be one of the reasons for the results. However, it is impossible to rule out the negative results contributed by the small sample cohort which comprised participants with non-mild hearing loss (especially in the [extremely] severe group; *n* = 54). This aspect, therefore, needs additional studies for verification. 

Our study had the following strengths and limitations. First, we adjusted the data to negate the effects of sex, age, health (common chronic diseases) and environmental exposure (lead and cadmium levels in blood as the internal dose of some external factors) as much as possible in the study design and analysis. Second, the RTL measured in peripheral blood cells may not accurately reflect the ongoing cellular aging in different tissues [7,9], and thus, the associations between hearing loss and RTL in the ear tissue might be stronger than those found in this study. Third, air-conduction audiometry was used in our study; although bone-conduction audiometry would have more accurately classified cases of sensorineural and conductive hearing loss, assessing these factors is rarely feasible in population studies [16]. 

## 5. Conclusions

In conclusion, to the best of our knowledge, this is the first case-control study to assess the association between hearing loss and telomere length at an epidemiological level. We found a strong and independent association between hearing loss and telomere length in the general population after adjustment for some potential confounders, particularly, mild hearing loss in females. Hence, telomere length in individuals who manifest the early symptoms of hearing loss may help prevent subsequent deterioration and hearing can be salvaged with appropriate management. Given the findings obtained in this study, further studies conducted among various ethnic cohorts are needed to verify the usefulness of telomere length as a biomarker to predict future risk of hearing loss, and to explore the possibility of a causal relationship between the two. 

## Figures and Tables

**Figure 1 ijerph-17-01881-f001:**
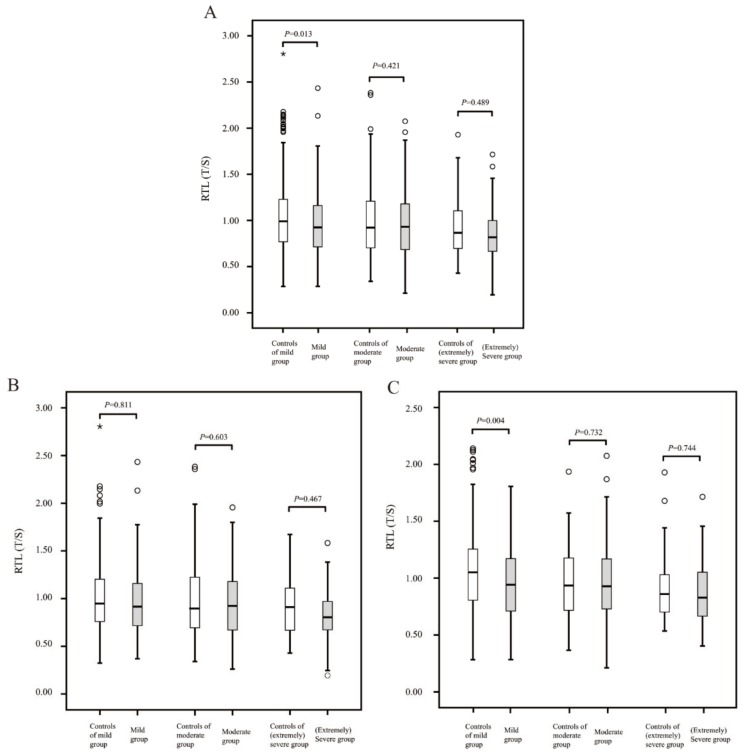
Box plot diagram of relative telomere length (RTL) in different populations ((**A**): total participants; (**B**): male participants and (**C**): female participants). Box plot explanation: upper horizontal line of box, 75th percentile; lower horizontal line of box, 25th percentile; horizontal bar within box, median; upper horizontal bar outside box, maximum value except for outliers; lower horizontal bar outside box, minimum value except for outliers; the hollow circles asterisks represent outliers; *p* < 0.05 indicates statistical significance between groups.

**Table 1 ijerph-17-01881-t001:** Sociodemographic Characteristics between Hearing Loss Cases and Controls.

Variables	Case (*n* = 817)	Control (*n* = 817)	*p*
Age, years	55.93 (8.99)	55.91 (9.03)	0.231
Sex (males), *n* (%)	411 (50.3)	411 (50.3)	1.000
RTL, median (P_25_–P_75_)	0.92 (0.71–1.15)	0.97 (0.75–1.22)	0.004
Hypertension, *n* (%)	268 (32.8)	148 (18.1)	<0.001
Hyperlipidemia, *n* (%)	64 (7.8)	31 (3.8)	0.001
Diabetes, *n* (%)	51 (6.2)	23 (2.8)	0.001
Hypercholesterolemia, *n* (%)	30 (3.7)	15 (1.8)	0.025
Lead (log-transformed), µg/dL	1.59 (0.16)	1.59 (0.16)	0.845
Cadmium (log-transformed), µg/L	0.51 (0.18)	0.50 (0.16)	0.043

Abbreviations: RTL, relative telomere length; Note: data of lead and cadmium are presented as mean (SD) based on the distribution of log-transformed. Wilcoxon signed-rank test for quantitative data with non-normal distribution (RTL). Paired *t*-test for quantitative data with normal distribution (age, lead and cadmium). McNemar test for qualitative data (sex, hypertension, hyperlipidemia, diabetes and hypercholesterolemia) in the paired design.

**Table 2 ijerph-17-01881-t002:** Conditional Logistic Regression Analysis of the Correlations of Hearing Loss.

Variables	Case, *n*	Control, *n*	OR (95% CI)	*p*-Trend
RTL				
Q1 (≤0.726)	186	186	1 (reference)	
Q2 (0.726–0.948)	198	198	0.76 (0.56–1.03)	
Q3 (0.948–1.200)	201	201	0.77 (0.57–1.05)	
Q4 (>1.200)	232	232	0.53 (0.38–0.74)	0.002
Hypertension				
No	549	669	1 (reference)	
Yes	268	148	2.10 (1.62–2.71)	<0.001
Hyperlipidemia				
No	753	786	1 (reference)	
Yes	64	31	1.44 (0.86–2.42)	0.162
Diabetes				
No	766	794	1 (reference)	
Yes	51	23	1.91 (1.08–3.38)	0.027
Hypercholesterolemia				
No	787	802	1 (reference)	
Yes	30	15	1.34 (0.62–2.90)	0.458
Cadmium	817	817	1.06 (1.01–1.12)	0.029

Abbreviations: RTL, relative telomere length; OR, odds ratio; CI, confidence interval; Q, quartile.

**Table 3 ijerph-17-01881-t003:** Odds Ratio (OR) (95% Confidence Interval (CI)) for Hearing Loss by Relative Telomere Length (RTL) Levels through Conditional Logistic Regression.

RTL Levels	Case/Control	Crude	Adjusted ^a^
		OR (95% CI)	OR (95% CI)
Per doubling of RTL	817/817	0.67 (0.53–0.83)	0.66 (0.52–0.83)
*p*-value		<0.001	<0.001
Per SD of RTL ^b^	817/817	0.80 (0.70–0.90)	0.79 (0.70–0.90)
*p*-value		<0.001	<0.001
RTL quartile			
Q1 (≤0.726)	186/186	1 (reference)	1 (reference)
Q2 (0.726–0.948)	198/198	0.81 (0.61–1.08)	0.76 (0.56–1.03)
Q3 (0.948–1.200)	201/201	0.80 (0.60–1.06)	0.77 (0.57–1.05)
Q4 (>1.200)	232/232	0.54 (0.39–0.74)	0.53 (0.38–0.74)
*p*-trend		0.002	0.002

Abbreviations: RTL, relative telomere length; OR, odds ratio; CI, confidence interval; Q, quartile; ^a^ Adjusted for hypertension, hyperlipidemia, diabetes, hypercholesterolemia and cadmium. ^b^ We modeled RTL as a continuous variable (per one standard deviation [SD] increase). One SD of RTL = 0.347.

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
