# Peer review of "Association between Leucocyte Telomere Length and Risk of Hearing Loss in the General Population: A Case-Control Study in Zhejiang Province, China"

_ijerph, 2020, doi:10.3390/ijerph17061881_

Round 1

Reviewer 1 Report

In this study by Zhang et. al the authors tried to establish a correlation between leucocyte telomere length and risk of hearing loss. As the author mentioned this kind of correlative studies have been done in past too. This study adds another layer in those findings with a region specificity. It will be better if the authors can add some other cohort data to make this more predictive for general population. In its current condition the manuscript can be accepted as the data and conclusion are in accordance with each other but this manuscript is lacks its importance as a novel work.

Author Response

Part A (Reviewer 1)

  1. Comment: In this study by Zhang et. al the authors tried to establish a correlation between leucocyte telomere length and risk of hearing loss. As the author mentioned this kind of correlative studies have been done in past too. This study adds another layer in those findings with a region specificity. It will be better if the authors can add some other cohort data to make this more predictive for general population. In its current condition the manuscript can be accepted as the data and conclusion are in accordance with each other but this manuscript is lacks its importance as a novel work.

       Answer: Thanks for the recognition and suggestion. And in our future research, we will further collect more information for a more comprehensive study of hearing loss. We also hope that more and more related research in this field will be published.

Reviewer 2 Report

I am glad to see the updated version. I read it again and I don't think I have more questions except one minor concern about the figure quality. According to the PDF file I read, the resolution and quality of the figures should be improved.

Author Response

Part B (Reviewer 2)

  1. Comment: I am glad to see the updated version. I read it again and I don't think I have more questions except one minor concern about the figure quality. According to the PDF file I read, the resolution and quality of the figures should be improved.

        Answer: Thank you very much for your recognition. And I am glad that you are interested in this article.

This manuscript is a resubmission of an earlier submission. The following is a list of the peer review reports and author responses from that submission.

Round 1

Reviewer 1 Report

Zhang H et al tried to link Leucocyte Telomere Length and hearing loss in their studies and demonstrated that LTL might be a potential predictive biomarker for early stage hearing loss. They chose the general population as target and set seven different healthcare centers for study, which is very important for discovering the relationship between telomere length and hearing loss and may help us to understand the mechanism of hearing loss via telomere biology. However, given the huge cost for samples collection and others at early stage, the authors should dig more detailed connections between telomere length and hearing loss.

Major concerns:

Even though this study set up to 7 centers, they focused on the population only in Zhejiang Province, China. The data might not be applicable throughout whole China, not to mention the whole human beings. Therefore, the authors should indicate this point in the title, just as a case-control study in Zhejiang Province, China The authors showed the association between RTL and hearing loss in table 1. Furthermore, the authors should separate the population to 2 groups based on gender. We really want to know if RTL influences hearing loss equally in male and female. It should also be noticed that the age range of the population for study is huge broad, from 21 to 89. In addition, we also knew that the hearing loss might mainly affect the aged people. Therefore, the population should be separated into 2 groups, like young people and old people. Then, the authors should show the influence of RTL on hearing loss in young and old population respectively. And if possible, it is better if the authors can find out a critical age which indicates whether RTL influence hearing loss based on age.

Other minor concerns:

Some descriptions should be modified. For example, in line 208, the authors demonstrated that “This study reports that the risk of hearing loss increases significantly with telomere shortening”. However, “telomere shortening” should be used in same individual, so “This study reports that the risk of hearing loss increases significantly in people with shorter telomeres” might be better. Check and correct the authorship. Check the format of references.

Author Response

Reviewer 1

Comment: Zhang H et al tried to link Leucocyte Telomere Length and hearing loss in their studies and demonstrated that LTL might be a potential predictive biomarker for early stage hearing loss. They chose the general population as target and set seven different healthcare centers for study, which is very important for discovering the relationship between telomere length and hearing loss and may help us to understand the mechanism of hearing loss via telomere biology. However, given the huge cost for samples collection and others at early stage, the authors should dig more detailed connections between telomere length and hearing loss.

Answer: Thanks for your recognition. We dug more detailed connections between telomere length and hearing loss based on your subsequent comments.

Comment: Even though this study set up to 7 centers, they focused on the population only in Zhejiang Province, China. The data might not be applicable throughout whole China, not to mention the whole human beings. Therefore, the authors should indicate this point in the title, just as a case-control study in Zhejiang Province, China.

Answer: Thanks for the suggestion. The content mentioned has been added to the title.

Comment: Furthermore, the authors should separate the population to 2 groups based on gender. We really want to know if RTL influences hearing loss equally in male and female.

Answer: Thank you for your advice. We have separated the population into four groups according to different genders and ages, to analyze the effect of RTL on hearing loss in different populations (Table S1). The correlation of RTL levels (per doubling and per SD increase) with hearing loss was found (Table S1). Besides, we also found statistically association of RTL with hearing loss after grouping by binary classification (the median as the critical value) other than RTL quartiles in women subgroup or young subgroup. We guess probably because the sample size of each subgroup would be too small which make the statistical power too weak after grouping by RTL quartiles. (Data not shown)..

Comment: It should also be noticed that the age range of the population for study is huge broad, from 21 to 89. In addition, we also knew that the hearing loss might mainly affect the aged people. Therefore, the population should be separated into 2 groups, like young people and old people. Then, the authors should show the influence of RTL on hearing loss in young and old population respectively. And if possible, it is better if the authors can find out a critical age which indicates whether RTL influence hearing loss based on age.

Answer: In Table S1, we separated the population into young group (≤ 60y) and elderly group (> 60y) to observe the effect of RTL on hearing loss in different age groups.

Comment: Some descriptions should be modified. For example, in line 208, the authors demonstrated that “This study reports that the risk of hearing loss increases significantly with telomere shortening”. However, “telomere shortening” should be used in same individual, so “This study reports that the risk of hearing loss increases significantly in people with shorter telomeres” might be better. Check and correct the authorship. Check the format of references.

Answer: Thanks for the suggestion and reminding. Related content has been modified in the manuscript. And all reference formats have been modified to MDPI format through endnote software.

Reviewer 2 Report

In this study by Zhang et al. the authors reported association between leucocyte telomere length and risk of hearing loss. Authors presented how telomere length can be associated with risk of hearing loss in general population.The authors showed that longer RTL is associated with a lower risk of developing hearing loss. They also used several different parameters like hypertension,diabetes, hyperlipidemia, hypercholesteromedia and metals like lead and cadmium to find out how these can effect the relationship between telomere length and loss of hearing. 

I found the study interesting, but it lacks several elements of a good manuscript specially in the experimental design and data prediction. Authors tried to present the study in detail but there are certain sections/observations in the study which raises some fundamental questions. The study is completely based on correlation of telomere length and hearing loss. There is no strong experimental evidence to further substantiate the authors claim about this correlation. Telomere length can depend on several different parameters including few which authors reported. In the result section the authors showed the case population has positive relationship with hypertension and diabetes. These chronic diseases also effect telomere length. This raises the question about authors observation. Can we use telomere length as a marker for hearing loss in patients without these chronic conditions? How does these conditions effect the relationship? Why or how shorter telomere are related with hearing loss? These fundamental questions needed to be answered in detail experiments before I can recommend this article for acceptance.

Author Response

Reviewer 2:

Comment: In this study by Zhang et al. the authors reported association between leucocyte telomere length and risk of hearing loss. Authors presented how telomere length can be associated with risk of hearing loss in general population. The authors showed that longer RTL is associated with a lower risk of developing hearing loss. They also used several different parameters like hypertension, diabetes, hyperlipidemia, hypercholesterolemia and metals like lead and cadmium to find out how these can affect the relationship between telomere length and loss of hearing.

Answer: Thanks for the affirmation and summary.

Comment: In the result section the authors showed the case population has positive relationship with hypertension and diabetes. These chronic diseases also effect telomere length. This raises the question about authors observation. Can we use telomere length as a marker for hearing loss in patients without these chronic conditions? How does these conditions effect the relationship?

Answer: As shown in the result section, these chronic diseases have been corrected for the analysis as confounding factors. After adjusting for these confounding factors, the results still showed an association between shorter telomere length and a higher risk of hearing loss. In addition, as shown in the table below, we also did the analysis to estimate the effect for hearing loss by RTL levels in people without these chronic conditions. The result showed that telomere length was also associated with hearing loss to some extent.

Table . OR (95% CI) for Hearing Loss by RTL Levels through Conditional Logistic Regression in people without chronic conditions a

Case/Control

Crude

Adjusted b

OR (95% CI)

OR (95% CI)

Per doubling of RTL

638/538

0.72 (0.55-0.95)

0.71 (0.54-0.93)

P-value

0.018

0.013

Per SD of RTL c

638/538

0.86 (0.75-0.99)

0.85 (0.74-0.98)

P-value

0.035

0.026

RTL quartile

638/538

Q1 (≤ 0.759)

1 (reference)

1 (reference)

Q2 (0.760-0.981)

0.84 (0.58-1.20)

0.82 (0.57-1.17)

Q3 (0.982-1.259)

0.65 (0.46-0.93)

0.64 (0.44-0.91)

Q4 (> 1.260)

0.72 (0.49-1.07)

0.70 (0.47-1.04)

P-trend

0.126

0.096

a Chronic conditions: hypertension, hyperlipidemia, diabetes, and hypercholesterolemia. b Adjusted for cadmium. c We modeled RTL as a continuous variable (per one standard deviation [SD] increase). One SD of RTL=0.374.

Comment: Why or how shorter telomere are related with hearing loss?

Answer: As stated in the 3rd paragraph of the discussion, shortened telomere can result in chromosomal instability and promote genomic lesions, which could lead to hearing loss. On the other hand, experiments have shown that oxidative damage caused by ROS plays a causal role in the development of hearing loss, and ROS can also lead to shortening of telomere. Of course, as one of analytical epidemiologic methods, case-control study can be used to investigate the association between the occurrence of a disease and an exposure suspected of causing that disease. Hence, this study can also provide a basis for further cohort studies and experimental studies on telomere length and hearing loss.

Reviewer 3 Report

In this study, Zhang et al. examined the correlation between hearing loss and relative telomere length. They found a correlation between shorter telomeres and increased hearing loss. The trends seem very subtle, but also seem to be supported by most of the data and statistical results. It is important that Zhang et al. normalized for age, since hearing loss is an age-associated condition. Some of the other conditions inventoried were also found to be correlated with RTL, but the authors say very little about these.

My main scientific concern is with the data and plot in Figure 1. The legend is inadequate since the lines within what is presumably the interquartile range for the “moderate” groups suggest the controls actually have a shorter RTL than the experimental group (with hearing loss). What is the explanation for this? Is the line within the box not the average, or is there actually a problem with the data and therefore also the generality of the conclusions? This aspect of Figure 1 was disconcertingly dissonant with the Results (lines 182–184) and conclusions of the paper.

Another issue with the manuscript is the Discussion. It needs logical refinement in the third paragraph in particular, where it rambles. In the third paragraph, I would suggest shifting the last sentence (lines 220–222) to becoming the topic for that paragraph and then expanding on possibly interesting correlative/causal relationships between telomere length/DNA damage and hearing loss. Presumably, the latter is the expertise of these researchers, so they must be well informed of issues of chromosomal instability and hearing.

Also, lines 190–192 did not make sense to me: what does it mean for the “association was not modified by the presence of hypertension, hyperlipidemia, …” ? This sentence should be rephrased to be more explicitly clear what the authors’ point is.

The table and figure legends need to be clearer. They do not explain all aspect of what the readers are looking at or define terms, statistics applied to specific numbers listed, etc. They should be expanded upon to be more forthcoming and allow clearer interpretation of the numbers and statistical tests used and shown, etc. This particularly applies to Figure 1, as stated above, but also Table 1; the legend does not even list what “P” is, yet legend mentions 3 different statistical tests performed — it is nearly unintelligible to me.

Finally, I think that the authors should provide a best-fit equation for Figure S1 (e.g., could be fit to a linear y = mx + b equation). This will allow comparison of how the data from this study compares with other analyses performed previously, as well as to be useful to researchers studying age-associated trends.

Author Response

Reviewer 3:

Comment: In this study, Zhang et al. examined the correlation between hearing loss and relative telomere length. They found a correlation between shorter telomeres and increased hearing loss. The trends seem very subtle, but also seem to be supported by most of the data and statistical results. It is important that Zhang et al. normalized for age, since hearing loss is an age-associated condition. Some of the other conditions inventoried were also found to be correlated with RTL, but the authors say very little about these.

Answer: Thanks for your recognition and summary. In addition, just as the title indicates, the focus of this study was RTL and hearing loss, hence, for the other conditions (such as hypertension, hyperlipidemia and diabetes), there were only corrected as confounding factors in the analysis, and not discussed too much.

Comment: My main scientific concern is with the data and plot in Figure 1. The legend is inadequate since the lines within what is presumably the interquartile range for the “moderate” groups suggest the controls actually have a shorter RTL than the experimental group (with hearing loss). What is the explanation for this? Is the line within the box not the average, or is there actually a problem with the data and therefore also the generality of the conclusions? This aspect of Figure 1 was disconcertingly dissonant with the Results (lines 182–184) and conclusions of the paper.

Answer: Since the Kolmogorov-Smirnov normality test showed the RTL was skewed distributed, the median (interquartile range) was used to describe the data. It can be seen from the box plot that the graph of “Moderate” group is lower than that of the control group as a whole (except for the median [the line within the box], RTL in the “Moderate” group was smaller than that in the control group, regardless of the minimum, maximum, upper quartile or lower quartile). In addition, we also put forward a certain degree of explanation and assumptions for the negative results of the non-mild hearing loss groups in the 4th paragraph.

Comment: Another issue with the manuscript is the Discussion. It needs logical refinement in the third paragraph in particular, where it rambles. In the third paragraph, I would suggest shifting the last sentence (lines 220–222) to becoming the topic for that paragraph and then expanding on possibly interesting correlative/causal relationships between telomere length/DNA damage and hearing loss. Presumably, the latter is the expertise of these researchers, so they must be well informed of issues of chromosomal instability and hearing.

Answer: Thanks for the reviewer’s kind advice. We have modified this paragraph to make it more logical.

Comment: Also, lines 190–192 did not make sense to me: what does it mean for the “association was not modified by the presence of hypertension, hyperlipidemia, …” ? This sentence should be rephrased to be more explicitly clear what the authors’ point is.

Answer: Thanks for your suggestion. What this sentence wants to express is that even after adjusting for these factors (hypertension, hyperlipidemia, diabetes, hypercholesterolemia, and cadmium in blood), there was still a significant association between RTL and hearing loss (results of columns “Crude” and “Adjusted” of Table 3). The sentence has been modified to make it clearer.

Comment: The table and figure legends need to be clearer. They do not explain all aspect of what the readers are looking at or define terms, statistics applied to specific numbers listed, etc. They should be expanded upon to be more forthcoming and allow clearer interpretation of the numbers and statistical tests used and shown, etc. This particularly applies to Figure 1, as stated above, but also Table 1; the legend does not even list what “P” is, yet legend mentions 3 different statistical tests performed — it is nearly unintelligible to me.

Answer: Thanks for the advice. The legends in Figure 1 and Table 1 have been modified to make the meaning clearer.

Comment: Finally, I think that the authors should provide a best-fit equation for Figure S1 (e.g., could be fit to a linear y = mx + b equation). This will allow comparison of how the data from this study compares with other analyses performed previously, as well as to be useful to researchers studying age-associated trends.

Answer: Thanks for your suggestion. The linear-fit equation has been added to Figure S1.

Round 2

Reviewer 1 Report

I am glad the authors have updated their results and have no more comments.

Reviewer 2 Report

In this revised manuscript the authors tried to answer the questions raised by the reviewer and it improved the quality of the manuscript. But I am still not convinced about the link between telomere length and hearing loss. The data which can conclusively tell that a link exist between the telomere length and hearing loss is still missing. The authors tried to answer the question about the reason behind this correlation by speculating DNA damage based loss of telomere length and hearing loss but without any experimental data it is very hard to conclude. Also, the region specific data limits the scientific significance of the work. Based on its current state I can not recommend this manuscript for publication without any experiment done to establish the speculative correlation.